# Breaking the Cloud Barrier: Local Deployment of Conversational LLMs for Real Time NPC Interaction in Video Games

*Abstract*—This paper explores the integration of a locally hosted Large Language Model (LLM), with Speech-to-Text (STT) and Text-to-Speech (TTS) systems within a real-time, open-world 3D video game produced on Unreal Engine 5. The research evaluates two primary aspects: the computational performance of AI processing on CPU versus GPU configurations and the impact of low-latency AI interactions on player engagement and immersion. Performance benchmarks are conducted to analyze system resource utilization, latency, and frame stability, while user experience assessments gauge the AI companion's effectiveness in enhancing gameplay dynamics. The results provide insights into the feasibility of offline-AI-driven NPCs, highlighting the trade-offs in computational efficiency and real-time interaction quality. This study provides a practical framework for real-time, offline AI driven game mechanics, enabling immersive and private interactions without cloud infrastructure.

*Index Terms*—Representation Learning, Multimodal Learning, Speech Processing, Natural Language Processing, Offline AI Systems, Real-Time Interaction

## I. Introduction and Related Work

In a world increasingly reliant on artificial intelligence, the intersection of storytelling, interactivity, and machine learning is beginning to reshape how we think about video games. In theory, these technologies enable non-playable characters (NPCs) in video games to speak, think, and respond with realism and spontaneity—an enticing promise for immersive storytelling and gameplay. However, in practice, this vision is often tethered to cloud-based platforms that introduce fundamental limitations for developers.

This paper presents a fully offline, locally hosted AI companion system for real-time use in high-fidelity 3D game environments. The key goals of this work are to eliminate reliance on cloud based services for AI driven NPC interactions, to evaluate and compare CPU vs GPU performance during live inference, and to demonstrate how character immersion can be achieved through fine- tuned language models and custom datasets.

Our work differs from prior work by offering a complete, end to end AI interaction pipeline, combining Speech To Text (STT), large language model (LLM) inference, and text-to-speech (TTS), that runs entirely on consumer grade hardware within Unreal Engine 5. Unlike systems that rely on persistent internet connections and external servers, our implementation preserves player privacy, eliminates usage based billing, and grants developers full control over AI behaviors and responses.

Most modern generative AI systems require remote infrastructure and subscription based APIs that introduce recurring costs, privacy concerns, and limitations on creative experimentation. These challenges are particularly restrictive for independent developers, hobbyists, and academic researchers. In contrast, our system promotes accessibility, transparency, and creative freedom by enabling developers to deploy real-time AI NPCs with no external dependencies.

Previous research has explored the computational dynamics of deploying AI in gaming environments. Kim et al. [1] demonstrated that GPUs significantly outperform CPUs in deep learning tasks due to their ability to handle parallelized matrix computations, a finding echoed by Singh et al. [2], who confirmed that GPUs outperform CPUs in both latency and throughput across deep learning models. These insights underline the importance of efficient hardware utilization for real-time AI systems, which this paper addresses by benchmarking CPU and GPU deployments during live inference.

In the domain of player immersion, Togelius and Levine [3] emphasized that AI-driven NPCs that adapt to player actions can foster emergent gameplay and stronger engagement. Eladhari and Sullivan [4] showed that neural network-driven NPCs create deeper emotional connections with players, particularly when responses are context-aware and grounded in real-time. For example, an AI companion might recall a player's earlier in-game decisions, such as choosing mercy over revenge, and later refer to it during a conversation, mirroring emotional continuity seen in games like *Red Dead Redemption 2* or *Mass Effect*. While these studies support the conceptual value of AI NPCs, they do not investigate implementation constraints such as system performance or offline viability.

Recent work has also highlighted a growing interest in localized AI architectures. Zhao et al. [5] compared AI-optimized FPGAs and GPUs for latency-sensitive applications, revealing that local inference can outperform cloud-based alternatives in real-time responsiveness. Kaur et al. [6] focused on the offline deployment of AI systems in gaming, finding improved reliability and reduced dependency on external servers. Still, both works stop short of exploring full system integration with 3D game engines or user speech interaction.

Park et al. [7] presented a breakthrough in AI simulation by introducing Generative Agents—LLM-powered virtual characters capable of memory retention, reflection, and emergent social behavior (i.e., actions and responses not explicitly pre-scripted, but learned through model interactions). These agents were deployed in sandbox environments and coordinated social events based on evolving internal states. Although

compelling, their system relies heavily on cloud-hosted models and is not built for direct, real-time player interaction or deployment in high-fidelity 3D environments. Unlike prior systems, this work delivers a fully self contained, speech based AI companion capable of real-time interaction within a high-fidelity game engine.

To the best of our knowledge, no prior work has demonstrated a fully offline AI companion architecture that integrates speech recognition, real-time language generation, and synthesized voice response within a live Unreal Engine 5 environment using only consumer-grade hardware. This paper addresses this gap by providing a replicable implementation pipeline that can be adopted by indie developers seeking local, affordable, and scalable AI companions.

The primary contributions of this paper are:

- A fully offline AI companion system combining STT, LLM inference, and TTS.
- Fine-tuning of a 1.1B parameter LLM (TinyLlama) on a custom dataset of over 1 million conversational exchanges formatted using ChatML.
- A custom-built TCP server that handles asynchronous speech-based AI interaction in real time with Unreal Engine 5.
- A performance analysis comparing CPU vs. GPU inference for conversational AI in gameplay settings.
- A demonstration of how character tone, personality, and immersion can be achieved through controlled dataset design and local AI modeling.

The remainder of this paper is organized as follows: Section II presents the methodology, including dataset construction, model training, quantization, and server integration. Section III evaluates the system through training loss metrics and hardware performance testing. Section IV concludes the paper with a discussion of limitations, design implications, and future directions.

## II. METHODOLOGY

This section outlines the design, optimization techniques, and implementation of integrating a locally hosted AI companion within Unreal Engine 5, focusing on low latency, real-time interactions. The methodology includes fine-tuning a small Large Language Model (LLM), TinyLlama-1.1B, using LoRA, quantizing the model into GGUF format for efficient inference, developing a TCP-based local server to handle AI processing, and seamless integration with Speech-to-Text (STT) and Text-to-Speech (TTS) systems. The training dataset was custom-built to reflect a specific character's personality and speech patterns, ensuring realistic, immersed AI-generated responses. Finally, we optimized the system through latency benchmarking, resource allocation analysis, and performance testing to ensure high response speed and minimal computational overhead in a fully offline environment.

### A. System Architecture

The AI system is designed to operate entirely offline with locally hosted architecture, ensuring low-latency responses without cloud dependencies. The architecture consists of the following key components:

- **Speech-to-Text (STT)**: Converts player speech into text using OpenAI's Whisper model.
- **LLM-Based Response Generation**: Processes the transcribed text through a fine-tuned TinyLlama model hosted locally to generate responses.
- **Text-to-Speech (TTS)**: Converts AI-generated responses into speech using Kokoro TTS.
- **TCP-Based Local Server**: Facilitates communication between the AI system and Unreal Engine 5.
- **MetaHuman Integration**: Synchronizes AI responses with lip-syn animations in Unreal Engine.

### B. Dataset Creation and Processing

The training dataset was created using a structured, character-driven approach to ensure realistic dialogue responses.

To fine-tune the AI companion to engage in authentic, emotionally grounded conversations, a large-scale custom dataset was created consisting of over one million paired conversational exchanges. Each row in the dataset represents a turn-based interaction between a user and the AI companion, with the first column titled "Input" containing the input prompt and the second column titled "Output" for the expected response of the model. The dataset follows a two-column format in CSV, which was ideal for pre-processing with Hugging Face datasets library and integrating with the training pipeline. This structure allows for direct alignment with Ollama's ChatML format. where inputs are implicitly interpreted as user messages and outputs are generated by the assistant module.

What sets this dataset apart is not only its size - totaling 1, 044, 995 rows - but also its tone, consistency, narrative depth, and format. Unlike standard instruction-based datasets, this dataset was entirely composed of organic, character-driven dialogue. The dataset was crafted to establish a consistent tone and vernacular for the AI companion. The conversations are causal, expressive, and culturally grounded, mirroring the speech patterns of the AI companion "Darnell". Each line in the dataset reads as if spoken by someone with lived experiences, often incorporating colloquial phrasing, humor, rhetorical questions, and deep personal reflection. For example, a sequence might begin with "Hey, what's up?" and be followed by a response like "Ayy, not much. What you got goin on?". In one scenario, a player might say "I'm feeling kinda off today," to which Darnell might respond, "Yeah? Want to talk it out or need a distraction? I got stories for days." This type of stylistic consistency reinforces the model's character voice and enhances believability in real-time interactions. The goal of fine tuning was to reflect Darnell's personality, not only through what was said, but also how it was said. The basis of a lot of characters comes from the tone and style of the words they speak and how they are phrased. Through Darnell, we wanted to convey this as a level on par with modern colloquial sayings to convey a character fit for our modern era.

The dataset was entirely custom-authored, allowing for precise control over tone and pacing. It avoids repetition and includes a wide linguistic variety to reduce overfitting and prevent generic robotic responses during inference. This is done by following a simple input-response pipeline in which the initial input is "Hi, how are you?" and the following output is "Ayy, not much. What you got goin on?". For the following row, the prior's rows output is inserted as the current row's input (input for current row = "Ayy, not much. What you got goin on?"). Rather than being a series of one-off question-answer pairs, the model is fine-tuned on a continuous dialogue flow. This approach trains the model to generate contextually coherent responses across entire conversations, rather than reacting in isolation. Importantly, the dataset avoids the use of special prompts, instead relying solely on the content of the input to drive context, reinforcing a cleaner, more scalable input-output format compatible with ChatML-based inferences.

From a technical perspective, the dataset was incorporated into the model by first uploading the dataset to the Hugging Face platform under the dataset title "KVN03/Darnell". This allows for reproducible access of the dataset across different training environments for different models. Tokenization and encoding was handled dynamically during the fine-tuning, and no additional prompt engineering, manual crafting of input instructions to guide model behavior,was required due to the dataset's clean formatting. This robust and stylistically faithful dataset served as a key foundation for the model's training and ultimately enable the AI companion to exhibit both high contextual awareness and a distinctive, engaging personality in-game.

### C. TCP-Based Local Server Integration

The real-time AI interaction system is enabled through a custom-built asynchronous TCP server written in Python, designed for low-latency communication between Unreal Engine 5 and the locally hosted AI processing pipeline. The server architecture is optimized to support real-time STT transcription, large language model inference making, and speech synthesis, all within an entirely offline environment. Unlike HTTP-based server systems, the use of raw TCP sockets ensures a persistent, lightweight connection with reduced overhead and lower latency, which is essential for synchronous interaction with AI companions in gameplay scenarios.

Upon initialization, the server adjusts its process priority and CPU affinity to maximize performance on multi-core systems such as those using AMD Ryzen 9 processors. The server is configured with high priority using the psutil library and bound to all logical cores to reduce context switching during execution. Once started, the server listens for connections on port 65432 using Python's socket library. The server was created to enable asynchronous handling of incoming data streams. This set-up allows the system to respond to Unreal Engine requests without blocking, ensuring minimal delay between player actions and AI feedback.

The interaction pipeline begins when Unreal Engine sends a signal via TCP, which the server interprets as a command to initiate STT processing. The server uses OpenAI's Whisper model in its lightweight "tiny" configuration, loaded once at runtime and executed either on CUDA enabled GPUs or the CPU based availability. The transcribe function uses Whisper's transcribe() method to process an input audio file (e.g., "Recording.wav") and return the corresponding text transcription. This transcription serves as the prompt for the next stage: response generation.

Once transcription is complete, the server optionally forwards the transcribed text to the fine-tuned TinyLlama model hosted locally through Ollama's AsyncClient. The server builds a ChatML-styled message object containing the user's input and sends it to the LLM, which returns a response in character. The use of asynchronous execution ensures that this model call does not block other server tasks. Moreover, by retaining a consistent interaction format, the system supports scalable expansion to multi-turn conversations and deeper contextual memory.

Following inference, the AI-generated response is passed into a Kokoro TTS pipeline. This module synthesizes speech from the response text using the American English male voice "am eric" configured for natural pacing and phonetic clarity. The resulting audio is saved to a local .wav file (e.g., "alecresponse.wav", which Unreal Engine can immediately access and play through the in-game NPC system. The final audio is lip-synced to a MetaHuman Character, creating the illusion of a sentient, voice aware companion.

Internally, the server uses asyncio's event loop to run all tasks non-blockingly, including file operations, LLM interaction, and audio generation. TCP-specific flags such as TCP NODELAY are also applied to minimize packet buffering and further reduce latency in transmission. The overall communication protocol is simple but effective: Unreal Engine sends a message (e.g., "1" to trigger STT), and the server returns either the AI generated speech or a controlled message such as "Done!!" after completing TTS synthesis.

This asynchronous design enables an event-driven server pipeline of a voice-based AI companion into an interactive game agent while preserving local execution, data privacy, and high performance. For instance, during an intense mission scene, the AI can respond dynamically to player stress cues, if the player hesitates or issues panicked voice commands, the NPC may say "We're not outta this yet, but I got your back!", reacting in real time with supportive tone and urgency. By optimizing system resources and leveraging efficient model architectures and libraries, the server enables lifelike AI behavior in real-time without relying on cloud infrastructure.

### D. Performance Considerations

To maintain optimal performance, the following optimizations were implemented:

- Asynchronous LLM processing via Ollama's AsyncClient for low latency execution.

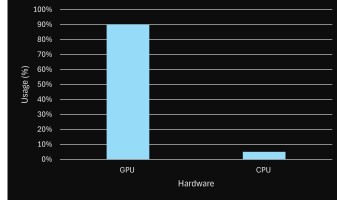

Fig. 1. Baseline Hardware Usage (%)

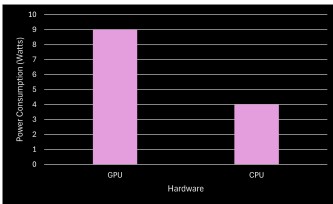

Fig. 2. Baseline Power Consumption (Watts)

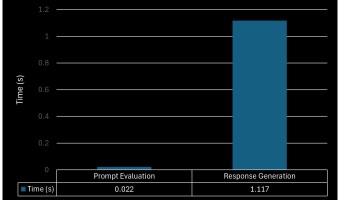

Fig. 3. Inference Time Breakdown – Baseline Model

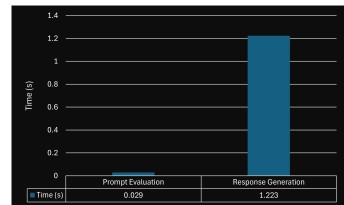

Fig. 4. Inference Time – Fine-Tuned Model

- Optimized server from a prior flask server to a TCP server for reliable connection and fast data transfer.
- Efficient memory management with chat history storage.

The preceding sections have outlined the design the design and implementation of the offline AI system, including model training, architectural choices, and server optimization. The next section evaluates the system's practical performance under real world conditions, highlighting inference latency, hardware utilization, and training outcomes.

## III. EXPERIMENTAL RESULTS

To evaluate the effectiveness and efficiency of the offline AI companion system, we collected data using the baseline TinyLlama-1.1B model and our fine-tuned tinyllama-darnell variant. Performance metrics were taken during live inference using Ollama's –verbose mode and system-level monitoring tools on a mid-range laptop with a 6GB VRAM GPU. These tests aimed to assess model responsiveness, hardware utilization, and training convergence behavior in a real-time gameplay context.

### A. Baseline Model

In its unfine-tuned state, TinyLlama displayed strong real-time inference performance. As shown in Figure 1, the GPU operated at approximately 90% utilization while the CPU load remained minimal at 5%, indicating that inference was primarily GPU-bound. Corresponding power consumption remained modest, with GPU and CPU drawing 9W and 4W respectively (Figure 2), demonstrating the feasibility of real-time deployment on mid-tier consumer hardware.

Inference timing was also profiled in detail. The system processed 914 prompt tokens in just 22 milliseconds, while generating 137 tokens of output took 1.117 seconds, resulting in an average generation speed of 122.65 tokens/second. As shown in Figure 3, prompt processing overhead was negligible, confirming that the generation phase remains the primary contributor to inference latency.

### B. Inference Timing Breakdown

To assess the effects of personalization, we fine-tuned TinyLlama using Low Rank Adaptation (LoRA) and observed resulting shifts in latency and output behavior. The following subsection evaluates inference time after tuning and training stability.

As shown in Figure 4, inference time slightly increased beyond 1.2 seconds, reflecting the added complexity required for generating personality-aligned responses. Despite this, the system remained within acceptable bounds for real-time interaction.

Training convergence was tracked across 12 000 steps, Figure 5 illustrates stable learning behavior, with a consistent decrease in training loss and a closely matched validation loss curve. The lack of overfitting suggests that the model generalizes well, successfully learning the stylistic and contextual patterns of the AI companion's personality.

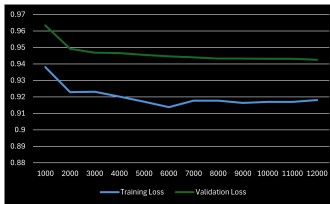

Fig. 5. Training Loss vs. Validation Loss (12k Steps)

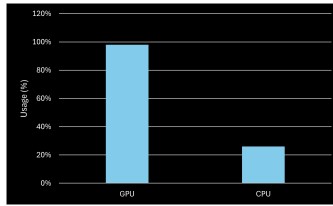

Fig. 6. Post-Tuning Hardware Usage (%)

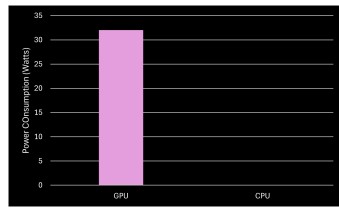

Fig. 7. Post-Tuning Power Consumption

*C. Inference After Fine-Tuning*

Fine-tuning did lead to increased hardware demand. GPU usage approached full saturation and power consumption rose to 32W (Figures 6 & 7). While this indicates a higher computational burden, the system remained fully operational without thermal throttling or dropped frames, reinforcing the system's viability under increased representational load.

## IV. Conclusion and Future Work

This study demonstrates the feasibility of deploying a fully offline AI companion system capable of speech-based interaction in a real-time 3D game environment. By combining fine-tuned LLM inference with Speech-To-Text and Text-To-Speech modules within a locally hosted TCP architecture, the system achieves immersive, privacy-preserving interactions without cloud dependencies. Experimental benchmarks confirm that both baseline and fine-tuned variants maintain real-time performance on consumer-grade hardware, with only modest increases in latency and power draw post-tuning.

Despite these promising results, several areas remain open for exploration. A primary direction is the deployment of larger language models, those with enhanced memory depth and contextual coherence, on high end GPUs with 12-24 GB of VRAM. This upgrade could enable longer conversational context, more expressive character output, and faster inference speeds under increased load. However, such advances bring new challenges, particularly in managing resource contention when deploying multiple models or agents on a single system.

Another critical area of future work is the development of distributed local inference, where multiple lightweight AI agents collaborate over a local network. For example, a team of NPCs in a squad based game could each run on different edge devices (i.e., LAN connected machines or modular AI boards), coordinating actions without requiring a central server or internet access. This architecture could unlock rich multi-character interactions while maintaining fuller user side control.

Further, we aim to expand spatial AI awareness by integrating perception modules that allow the AI to observe, interpret, and respond to in-game environments dynamically. This includes proximity based dialogue, real-time analysis, and visual event triggers via camera feeds or environmental sensors. A companion that notices, remembers, and reacts to its surroundings transforms from a reactive chatbot into a narratively intelligent presence grounded in the world it inhabits.

Beyond technical enhancements, this work advocates for self sovereign AI systems, those that uphold player privacy, resist platform dependency, and empower developers through full stack ownership. Our findings offer a replicable blueprint for indie developers, researchers, and game studios interested in building expressive, responsive NPCs untethered from remote APIs or subscription models.

With ongoing improvements in local hardware acceleration and creative dataset design, the next generation of AI companions will not only rival but potentially surpass the interactivity and richness of cloud-based systems, ushering in a new era of ethically grounded, locally intelligent game AI.

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
