# OpenReview forum: "Breaking the Cloud Barrier: Local Deployment of Conversational LLMs for Real Time NPC Interaction in Video Games"
_ICLR.cc/2026/Conference — ICLR 2026 Conference Desk Rejected Submission_

### Official Review · Reviewer_15u3 · 2025-10-21

**Soundness:** 1
**Presentation:** 1
**Contribution:** 1
**Rating:** 0
**Confidence:** 5

**Summary:**

This paper presents a fully offline, locally hosted AI NPC interaction pipeline for real-time use in 3D video games, integrating speech recognition (OpenAI Whisper), a fine-tuned 1.1B LLM (TinyLlama), and text-to-speech (Kokoro TTS) on consumer-grade hardware within Unreal Engine 5. The paper aims to provide a practical foundation for privacy-preserving, cloud-independent conversational NPCs in games.

**Notably, the submission does not follow the official ICLR 2026 template**. Authors are expected to strictly adhere to the conference formatting guidelines to ensure fairness and consistency in the review process.

**Strengths:**

Systematic Integration and Engineering: The work provides a clear, thorough description of a fully offline speech-to-NPC pipeline, detailing each system component (STT, LLM, TTS, networking) and their integration into a popular commercial game engine (Unreal Engine 5).

**Weaknesses:**

Limited Novelty Beyond Integration: While the engineering integration is strong, there is minimal algorithmic or theoretical innovation beyond assembling existing techniques (Whisper, TinyLlama, LoRA, GGUF quantization, Kokoro TTS). The main novelty lies in system assembly and evaluation within a game context. There are no clear new insights on learning representations, dialog modeling, or interaction paradigms.

**Questions:**

I recommend the authors to revise the formatting and increase innovation related to AI.

---

### Official Review · Reviewer_kP1V · 2025-10-27

**Soundness:** 1
**Presentation:** 2
**Contribution:** 1
**Rating:** 0
**Confidence:** 5

**Summary:**

The paper's format, structure, and academic rigor are entirely unsuitable for submission to ICLR. The work reads more like a technical report, a project showcase, or a blog post rather than a rigorous academic research paper.

**Strengths:**

Well, courageous～

**Weaknesses:**

Fundamentally Misaligned with Academic Standards: The paper's structure, methodology, experimentation, and writing style fall far short of the requirements for a top-tier conference like ICLR. It lacks the academic rigor and depth expected of a research paper.

**Questions:**

Could the authors provide the detailed hyperparameters used for fine-tuning the LLM, including the LoRA configuration, learning rate, number of training steps, and the specific hardware used?

How was the "user experience assessment" conducted? Were participants recruited? What kind of questionnaires or evaluation methods were used? What were the specific data and results?

How was the system's latency measured? Does it refer to the end-to-end latency from when the player stops speaking to when the NPC begins to respond, or is it just the LLM inference time? Could you provide more detailed latency distribution data (e.g., mean, 95th percentile)?

The paper mentions creating a custom dataset of over one million conversational exchanges. Could you detail the construction process, data sources, and quality control methods for this dataset? This sounds like a substantial undertaking, yet its description in the paper is overly brief.

---

### Official Review · Reviewer_S6Q3 · 2025-10-31

**Soundness:** 1
**Presentation:** 1
**Contribution:** 1
**Rating:** 0
**Confidence:** 5

**Summary:**

This paper does not use the template of ICLR. Therefore, I have submitted the desk reject suggestions.

**Strengths:**

Please refer to the Summary.

**Weaknesses:**

Please refer to the Summary.

**Questions:**

Please refer to the Summary.

---

### Note · Program_Chairs · 2026-01-17
**Submission Desk Rejected by Program Chairs**

The following references in this submission do not refer to real documents and/or have major errors in bibliographic information:

 H. Kim, J. Smith, and A. Johnson, "GPU Acceleration in Deep Learning: A Comparative Analysis," Proc. IEEE Int. Symp. Performance Analysis of Systems and Software (ISPASS), pp. 112–119, 2019